# Predictors of Mortality in Patients with Infections Due to Carbapenem-Resistant Gram-Negative Bacteria

**DOI:** 10.3390/antibiotics12071130

**Published:** 2023-06-29

**Authors:** Hector Orlando Rivera-Villegas, Bernardo Alfonso Martinez-Guerra, Rosalia Garcia-Couturier, Luis Fernando Xancal-Salvador, Veronica Esteban-Kenel, Ricardo Antonio Jaimes-Aquino, Miguel Mendoza-Rojas, Axel Cervantes-Sánchez, Steven Méndez-Ramos, Jorge Eduardo Alonso-Montoya, Diana Munguia-Ramos, Karla Maria Tamez-Torres, Carla Marina Roman-Montes, Sandra Rajme-Lopez, Areli Martínez-Gamboa, Miriam Bobadilla-del-Valle, Maria Fernanda Gonzalez-Lara, Jose Sifuentes-Osornio, Alfredo Ponce-de-Leon

**Affiliations:** 1Department of Infectious Diseases, Instituto Nacional de Ciencias Médicas y Nutrición Salvador Zubirán, 15 Vasco de Quiroga, Belisario Domínguez Secc 16, Tlalpan, Mexico City 14080, Mexico; hector.riverav@incmnsz.mx (H.O.R.-V.); zs16012181@estudiantes.uv.mx (R.G.-C.); dmunguiar8@hotmail.com (D.M.-R.); karla.tamezt@incmnsz.mx (K.M.T.-T.); carla.romanm@incmnsz.mx (C.M.R.-M.); sandra.rajmel@incmnsz.mx (S.R.-L.); 2Clinical Microbiology Laboratory, Department of Infectious Diseases, Instituto Nacional de Ciencias Médicas y Nutrición Salvador Zubirán, 15 Vasco de Quiroga, Belisario Domínguez Secc 16, Tlalpan, Mexico City 14080, Mexico; fernando.xancals@incmnsz.mx (L.F.X.-S.); veronica.estebank@incmnsz.mx (V.E.-K.); ricardo.jaimesa@incmnsz.mx (R.A.J.-A.); miguel.mendozar@incmnsz.mx (M.M.-R.); axel.cervantess@incmnsz.mx (A.C.-S.); steven.mendezr@incmnsz.mx (S.M.-R.); rosa.martinezg@incmnsz.mx (A.M.-G.); miriam.bobadillav@incmnsz.mx (M.B.-d.-V.); fernanda.gonzalezl@incmnsz.mx (M.F.G.-L.); 3Department of Medicine and Health Sciences, Universidad de Sonora, Blvd. Luis Encinas J, Calle Av. Rosales & Centro, Hermosillo 83000, Mexico; a218200204@unison.mx; 4General Direction, Instituto Nacional de Ciencias Médicas y Nutrición Salvador Zubirán, 15 Vasco de Quiroga, Belisario Domínguez Secc 16, Tlalpan, Mexico City 14080, Mexico; jose.sifuenteso@incmnsz.mx

**Keywords:** carbapenem resistance, carbapenemase, mortality, antibiogram

## Abstract

Introduction: Infections caused by carbapenem-resistant Gram-negative bacteria (CR-GNB) are a significant cause of mortality and represent a serious challenge to health systems. The early identification of mortality predictors could guide appropriate treatment and follow-up. We aimed to identify the factors associated with 90-day all-cause mortality in patients with CR-GNB infections. Methods: We conducted a cohort study from 1 January 2019 to 30 April 2022. The primary outcome was death from any cause during the first 90 days after the date of the first CR-GNB-positive culture. Secondary outcomes included infection relapse, invasive mechanical ventilation during follow-up, need for additional source control, acute kidney injury, *Clostridioides difficile* infection, and all-cause hospital admission after initial discharge. Bivariate and multivariate Cox-proportional hazards models were constructed to identify the factors independently associated with 90-day all-cause mortality. Results: A total of 225 patients with CR-GNB infections were included. Death occurred in 76 (34%) cases. The most-reported comorbidities were immunosuppression (43%), arterial hypertension (35%), and COVID-19 (25%). The median length of stay in survivors was 18 days (IQR 10–34). Mechanical ventilation and ICU admission after diagnosis occurred in 8% and 11% of cases, respectively. Both infection relapse and rehospitalisation occurred in 18% of cases. *C. difficile* infection was diagnosed in 4% of cases. Acute kidney injury was documented in 22% of patients. Mechanical ventilation after diagnosis, ICU admission after diagnosis, and acute kidney injury in the first ten days of appropriate treatment were more frequently reported among non-survivors. In the multivariate analysis, age (HR 1.19 (95%CI 1.00–1.83)), immunosuppression (HR 1.84 (95%CI 1.06–3.18)), and septic shock at diagnosis (HR 2.40 (95% 1.41–4.08)) had an independent association with death during the first 90 days after the CR-GNB infection diagnosis. Receiving antibiogram-guided appropriate treatment was independently associated with a lower risk of death (HR 0.25 (95%CI 0.14–0.46)). Conclusions: The presence of advanced age, immunosuppression, septic shock at diagnosis, and inappropriate treatment are associated with higher 90-day all-cause mortality in hospitalised patients with infections due to CR-GNB. Recognition of the risk factors for adverse outcomes could further assist in patient care and the design of interventional studies that address the severe and widespread problem that is carbapenem resistance.

## 1. Introduction

Infections caused by carbapenem-resistant Gram-negative bacteria (CR-GNB) are a significant cause of morbidity and mortality. The World Health Organization listed broad-spectrum antimicrobial resistance (AMR) as one of the top-ten threats to global health [1]. Because carbapenems are an effective therapeutic alternative for multidrug-resistant (MDR) organisms, widespread carbapenem resistance represents a serious challenge to health systems. A progressive global increase in the incidence of CR-GNB in the last two decades has been identified, with reports of hospital outbreaks worldwide [2,3,4]. The latter has been associated with increased mortality and burden of disease [5]. A mortality of up to 50% has been reported among patients with bacteraemia due to carbapenem-resistant Enterobacterales (CRE) [6]. Additionally, the presence of carbapenemases has been independently associated with increased mortality [7]. In retrospective studies, the use of inappropriate antibiotic therapy, a higher APACHE score, a Charlson comorbidity index >3, advanced age, chemotherapy-induced neutropenia, and septic shock have been associated with increased mortality among patients with infections due to CR-GNB [8,9]. Based on estimates from a model conducted in the US in 2017, the average hospital cost of a single episode of an infection due to CR-GNB ranges between USD 22,484 and 66,031 [8].

It is essential to improve diagnostic methods and treatment strategies for patients with infections due to carbapenem-resistant organisms (CROs) [9]. As CROs disseminate, the factors associated with mortality could vary between regions [10]. In our region, scarce evidence exists regarding prognostic factors in patients with infections due to CROs. Unidentified factors could contribute to an excess in mortality, especially in regions with limited availability of first-line antibiotics. The early identification of mortality predictors could guide treatment and follow-up. Additionally, the recognition of factors associated with a worse prognosis may contribute to the design of research protocols focusing on high-risk patients.

We conducted a retrospective cohort study to identify the risk factors associated with 90-day all-cause mortality in patients with infections due to CR-GNB.

## 2. Methods

We conducted a retrospective cohort study in a tertiary care centre in Mexico City, which included all hospitalised patients aged 18 or older and diagnosed with any culture-proven infection due to carbapenem-resistant Enterobacterales or non-fermenting GNB during the period from 1 January 2019 to 30 April 2022. A CRO was considered when non-susceptibility to at least one carbapenem was documented. Patients with community- or hospital-acquired infections were included. Community- and hospital-acquired infections were defined as previously recommended [11]. All samples sent to the clinical microbiology laboratory were screened, regardless of their origin. We excluded patients in whom the CR-GNB isolate was considered not clinically significant according to the attending team’s criterion.

The isolates were identified using matrix-assisted laser desorption/ionisation time-of-flight (MALDI-TOF) (Brucker Daltonics, Bremen, Germany). Susceptibility was obtained using VITEK-2^®^ (BioMérieux, Marcy-L’Étoile, France). If a CRO was detected, further phenotypic tests such as modified and EDTA-modified carbapenem inactivation methods (mCIM/eCIM) and broth microdilution were performed according to the Clinical & Laboratory Standards Institute [12]. To identify distinct carbapenemases, an in-house polymerase chain reaction (PCR) using previously validated primers (Integrated DNA Technologies, Coralville, IA, USA) was performed [13].

Demographics, clinical, treatment, laboratory, and microbiology data were collected from the electronic medical records. Immunosuppression was considered when immunosuppressive medication (e.g., chemotherapeutic agents, drugs to prevent graft rejection, methotrexate, ≥10 mg of prednisone or equivalent for the last 14 days, monoclonal antibodies) or comorbidities (e.g., solid malignant tumours, hematologic malignancy, solid organ transplant, human immunodeficiency virus infection, and connective tissue disorders) were present. Screening for CRO colonisation is not a routine practice in our centre. The treatment data included antimicrobial prescription and source control procedures. Antibiogram-guided appropriate treatment was considered when a combination therapy with at least one active antimicrobial, as reported by antibiogram, was used. For carbapenemase-harbouring isolates, carbapenem monotherapy was considered inappropriate. Data regarding the outcomes were also retrieved from the medical records. The participants were followed up for 90 days after the date of the first CR-GNB-positive culture.

The primary outcome was death from any cause during the first 90 days after the date of the first CR-GNB-positive culture. An infection-related death was considered when patients died because of direct infectious complications (e.g., septic shock, pneumonia, source control surgical complications) or in cases of persistent signs of infection such as fever, persistently elevated leucocyte counts, C-reactive protein or procalcitonin, or persistent positive blood cultures. The secondary outcomes included death from any cause during the first 30 days after the date of the first CR-GNB-positive culture, length of stay, infection relapse, defined by the presence of clinical deterioration as assessed by the attending clinical team in addition to microbiologic confirmation of the index isolate, invasive mechanical ventilation (IMV) after the diagnosis of infection, need for additional source control, acute kidney injury (e.g., creatinine increase of >0.3 mg/dL of serum creatinine when compared to baseline) during the first ten days after antimicrobial treatment initiation, *Clostridioides difficile* infection, and all-cause hospital admission after initial discharge within the first 90 days after the date of the first CR-GNB-positive culture.

Considering a primary outcome probability of 50% [2,3], a mean absolute percentage error of 10%, and the identification of up to 10 potential predictors, we calculated a sample size of at least 165 patients [14]. Descriptive statistics were reported using mean and standard deviation or median and interquartile range (IQR) according to the variables’ distribution determined by the Shapiro–Wilk test. Chi-square, Fisher’s exact test, T-test for independent samples, and rank sum tests were used for comparisons between groups. To identify the factors associated with mortality, a bivariate analysis was performed to calculate the hazard ratio (HR) and 95% confidence intervals (95%CI). A multivariate Cox-proportional hazards model was constructed to identify the factors independently associated with 90-day all-cause mortality. The model was constructed using variables of biological importance according to previous reports. Variables with interactions as assessed by Mantel–Haenszel Chi were not included in the model (see Appendix A for detailed information). Missing data were not replaced. A p-value of 0.05 was considered statistically significant. The analysis was carried out using STATA V15 (Houston, TX, USA). Because of the study’s retrospective nature, the informed consent requirement was waived. The study, including the waived informed consent, was approved by the Institutional Review Board (ref. 4022). All personal data were protected according to national and international standards.

## 3. Results

A total of 288 patients with CRO isolates were screened during the study period. A total of 225 patients were included. All-cause 90-day mortality occurred in 76/225 (34%) cases of which 51/76 (67%) died from infectious causes, and 25/76 (33%) died from non-infectious causes (Figure 1).

A total of 145/225 (64%) patients were male, the median age was 54 years (IQR 40–66), and the time from admission to diagnosis was 13 days (IQR 4–28). The most frequent causes of hospital admission were bacterial infection in 125/225 (56%), COVID-19 in 45/225 (20%), and neoplastic diseases in 24/225 (11%) cases. A total of 81/225 cases (36%) were diagnosed in the intensive care unit (ICU). Respiratory tract, intraabdominal infections, and primary bloodstream infections occurred in 88/225 (39%), 85/225 (38%), and 19/225 (8.4%) cases, respectively. More than one episode occurred in 18/225 (8%) patients. The most reported comorbidities were immunosuppression in 96/225 (43%), arterial hypertension in 78/225 (35%), and COVID-19 in 56/225 (25%) cases. A Charlson comorbidity index greater than 3 was calculated in 88/225 (39%) patients. At the time of diagnosis, 67/225 (30%) patients were on IMV, while 62/225 (28%) had septic shock. Greater age, intensive care unit (ICU) at the time of diagnosis, respiratory and skin and soft tissue infections, higher Charlson comorbidity index, heart disease, immunosuppression, IMV use at diagnosis, and septic shock at diagnosis were more frequently observed among non-survivors (Table 1).

Of 85 patients with intraabdominal infection, source control was performed in 65 patients (76%). In patients with intraabdominal infections, no difference in the frequency of death was observed between those in whom source control was obtained and those in whom source control was not obtained (10/65, 15% vs. 6/20, 30%, *p* = 0.14). In the previous 180 days before diagnosis, the use of antibiotics, intravascular devices, use of healthcare services, and ICU stay were reported in 204/225 (91%), 173/225 (77%), 140/225 (62%), and 94/225 (42%) patients, respectively. A total of 195/225 (87%) patients were receiving antibiotic treatment at the time of diagnosis. Chemotherapy in the previous 180 days was reported in 48/225 (21%) and was more frequent in non-survivors (34.2% vs. 14.9%, *p* < 0.01). Median haemoglobin, leukocyte count, total lymphocytes count, platelet count, glucose, albumin, and C-reactive protein at diagnosis were 9.1 g/dL, 9.1 cells ×10^3^/μL, 0.6 cells ×10^3^/μL, 221 cells ×10^3^/μL, 116 mg/dL, 2.5 g/dL, and 13.6 mg/dL, respectively. Non-survivors had a higher median glucose (136 vs. 108 mg/dL, *p* < 0.01) and C-reactive protein (16 vs. 12.4 mg/dL, *p* < 0.01) and lower haemoglobin (8.4 vs. 9.5 g/dL, *p* < 0.01), platelet counts (149 vs. 278 x10^3^/μL, *p* < 0.01), and albumin levels (2.4 vs. 2.7 g/L, *p* < 0.01) at diagnosis. Appendix A summarise the risk factors for CR-GNB and the laboratory values at the time of diagnosis.

*Pseudomonas aeruginosa* (95 isolates), *Escherichia coli* (68 isolates), and *Klebisella pneumoniae* (21 isolates) were the most frequently recovered organisms. Bloodstream and intraabdominal infections were mainly caused by *E. coli* (10/19 and 37/85, respectively), whereas *P. aeruginosa* was present in most respiratory and urinary tract infections (46/88 and 11/28, respectively). Among *Enterobacteriaceae*, most *E. coli* and *K. pneumoniae* strains showed resistance to third-generation cephalosporins (56/59 and 20/21, respectively), piperacillin/tazobactam (56/59 and 19/19, respectively), and quinolones (61/66 and 19/21, respectively). In *Enterobacter* complex species, resistance to piperacillin/tazobactam (6/13) and quinolones (3/15) was less frequent. Resistance to amikacin was observed in six isolates (three *E. coli* and three *K. pneumoniae*). Except for two isolates (one *E. coli* and one *K. aerogenes*), all strains tested were susceptible to tigecycline. Among non-fermenting GNB, the antibiotics that presented a lower frequency of non-susceptibility in *P. aeruginosa* were colistin, amikacin, and ciprofloxacin in 4/81, 20/93, and 32/91, respectively. A lack of susceptibility to piperacillin/tazobactam and ceftazidime was observed in 39/86 and 35/93, respectively. Regarding *A. baumanii*, 2/6 strains were non-susceptible to tigecycline. The presence of carbapenemases was confirmed in most strains of *E. coli* (48/63), *K. pneumoniae* (18/20), *Raoultella sp.* (7/7) and *Citrobacter freundii* (1/1), with a total of 79 positive results out of the 126 performed phenotypic tests. Using PCR, NDM and OXA-48 were the most frequently found carbapenemases, in 26/88 and 23/88 cases, respectively. Appendix A summarise the microbiological data. Antibiogram-guided appropriate treatment was more frequent in survivors (139/149 (93%) vs. 57/76 (75%), *p* < 0.01) and is summarised in Appendix A. Of note, 63/196 (32%) patients received antibiogram-guided appropriate combination therapy. The most frequently reported antibiogram-guided appropriate treatment included amikacin in 35/196 (18%), tigecycline in 35/196 (18%) and piperacillin-tazobactam in 31/196 (16%) patients. Overall, the median antibiotic duration antibiotic was 10 days (IQR 7–15) (Appendix A).

Death from any cause during the first 30 days after the date of the first CR-GNB-positive culture occurred in 54/225 (24%) patients. The median length of stay in survivors was 18 days (IQR 10–34). The median time from diagnosis to death was 13 days (IQR 4–38). Mechanical ventilation and ICU admission after diagnosis occurred in 19/225 (8%) and 25/225 (11%) patients, respectively. Both infection relapse and rehospitalisation for any cause occurred in 41/225 (18%) cases. A *C. difficile* infection was diagnosed in 9/225 (4%) patients. Acute kidney injury was documented in 40 of 181 (22%) patients with available data. Mechanical ventilation after diagnosis, ICU admission after diagnosis, and acute kidney injury in the first ten days of appropriate treatment were more frequently reported among non-survivors (Table 2).

In the bivariate analysis, increasing age, respiratory tract infections, bone and soft tissue infections, Charlson comorbidity index >3, heart disease, immunosuppression, renal replacement therapy, IMV at diagnosis, septic shock at diagnosis, chemotherapy prescription 180 days prior to diagnosis, ICU at diagnosis, lower levels of haemoglobin, leukocytes, platelets, and albumin at diagnosis, higher levels of glucose, creatinine and C-reactive protein at diagnosis, acute kidney injury during treatment, IMV after diagnosis, and ICU stay after diagnosis were associated with higher HR for death during the first 90 days after the CR-GNB infection diagnosis. Intraabdominal infections and antibiogram-guided appropriate antibiotic treatment were associated with lower HR for death during the first 90 days after the CR-GNB infection diagnosis. Appendix A summarises the non-adjusted bivariate analysis.

In the multivariate analysis (Table 3), age (HR 1.19 (95%CI 1.00–1.83)), immunosuppression (HR 1.84 (95%CI 1.06–3.18)), and septic shock at diagnosis (HR 2.40 (95% 1.41–4.08)) had an independent association with 90-day all-cause mortality. In contrast, antibiogram-guided appropriate treatment was independently associated with a lower risk of death (HR 0.25 (95%CI 0.14–0.46)). Figure 2, Figure 3, Figure 4 and Figure 5 depict the unadjusted cumulative probability of survival.

## 4. Discussion

In our study, older age, immunosuppression, and septic shock at diagnosis of the infection were independently associated with a higher risk of death in patients with infections due to CR-GNB. In contrast, receiving antibiogram-guided appropriate treatment was independently associated with a lower risk of death. Our results could be explained by the fact that advanced age and immunosuppression pose a higher risk for increasingly severe forms of infection; septic shock remains the most clinically severe form of an infectious disease. Our results are in accordance with previous reports. In Scotland, Zhao et al. [15] reported that age >60 years and organ failure were associated with higher 30-day mortality in patients with infections due to carbapenemase-producing organisms. In the study carried out by Gualtero et al. [16], the presence of septic shock was associated with increased 30-day mortality in patients with infections due to CRE. Similarly, immunosuppression was associated with higher mortality in patients with CRE infections in a study conducted in Israel [17]. A higher mortality has been associated with inappropriate treatment in the context of CR organisms [18,19,20,21]. Additionally, clinical trials have reported an impact on mortality based on the type of treatment administered [22,23,24]. Inappropriate and second-line treatment options have been associated with unfavourable clinical response and toxicity [25]. Our mortality rate remains within the range previously reported [26].

Respiratory tract and intraabdominal infections contributed to most of the CR-GNB infections. According to our results, the site of infection was not associated with a higher risk of death. To our knowledge, there are no studies comparing mortality between different sites of infection in the setting of CR. A high prevalence of chronic degenerative diseases and immunosuppressive conditions was found in the studied population, which is explained by the characteristics inherent to the population treated in our institution. Unlike previous reports [15,17], we found no isolate-driven differences in mortality. The latter could be because the low number of *Acinetobacter* sp. isolates. Of note, isolate frequency varied according to the type of infection. Higher frequencies of *P. aeruginosa* and *Enterobacteriaceae* in respiratory tract and intraabdominal infections, respectively, were expected. Although some studies correlate the presence of acute kidney injury with higher mortality [27,28], evidence in patients with infections due to CR-GNB infections is scarce.

Among the limitations of our study, its retrospective nature must be considered. Regardless of the retrospective design, systematic data gathering was undertaken, and missing information was scarce and accounted for during the statistical analysis. The single-centre nature of our study could limit the applicability of the results. In our study, the effect of previous CRE colonisation was not measured. Also, we were unable to report complete genotypic profiles; therefore, specific carbapenem resistance mechanisms’ contributions to mortality could not be analysed. In most of the reported studies, complete molecular data on resistance mechanisms are unavailable, so their absence does not necessarily imply a major bias. Although including different infectious syndromes and isolates in the same model could limit our results, the fact that none of them were independently associated with a higher risk of death may provide further strength to our results, being that we describe factors that are associated with worse outcomes regardless of specific clinical scenarios. Nevertheless, sample heterogeneity regarding isolates and clinical scenarios must be considered when interpreting our results. Previous studies have suggested that, rather than carbapenem resistance, difficult-to-treat resistance is associated with mortality [29]. Of note, our study focuses predominantly on CR isolates. Additionally, although antibiogram-guided appropriate treatment was independently associated with lower mortality, the fact that some patients not receiving appropriate treatment died within the first days after diagnosis must be considered. Given that the study was not initially designed accordingly, the effect of different antibiotics and antibiotic combinations could not be measured. The significant sample size and extensive data gathering represent strengths in our study. Of note, we chose to study outcomes at 90 days after diagnosis to ensure a better understanding of the long-term impact of infections due to CRO. The latter enabled us to document outcomes that could otherwise be unnoticed. Additionally, patients with infections due to CRO tend to have a higher frequency of non-infectious comorbidities that could impact the outcomes after a 30-day follow-up.

## 5. Conclusions

Prospective studies are needed to support our findings. Also, more information regarding resistance mechanisms and mortality is warranted. Because antibiogram-guided appropriate treatment was associated with a lower risk of death, we believe that our study highlights the fact that implementing fast and reliable diagnostic methods is necessary to reduce the time to diagnosis and, hence, appropriate treatment. An accurate diagnosis may contribute to reducing mortality attributed to suboptimal treatment. The recognition of prognostic factors is essential to implement adequate preventive and therapeutic care to improve patient outcomes. Additionally, the recognition of risk factors for adverse outcomes could further assist in patient care and the design of interventional studies that address the severe and widespread problem that is carbapenem resistance.

Advanced age, immunosuppression, septic shock at diagnosis, and inappropriate treatment are associated with higher 90-day all-cause mortality in hospitalised patients with infections due to CR-GNB. 

## Figures and Tables

**Figure 1 antibiotics-12-01130-f001:**
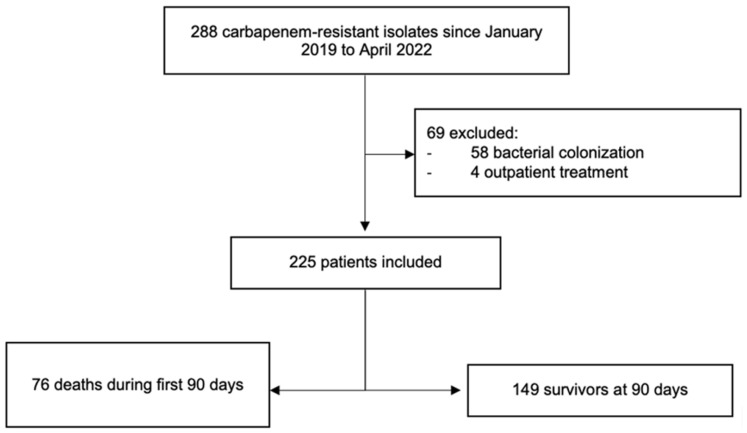
Flow diagram of patient inclusion.

**Figure 2 antibiotics-12-01130-f002:**
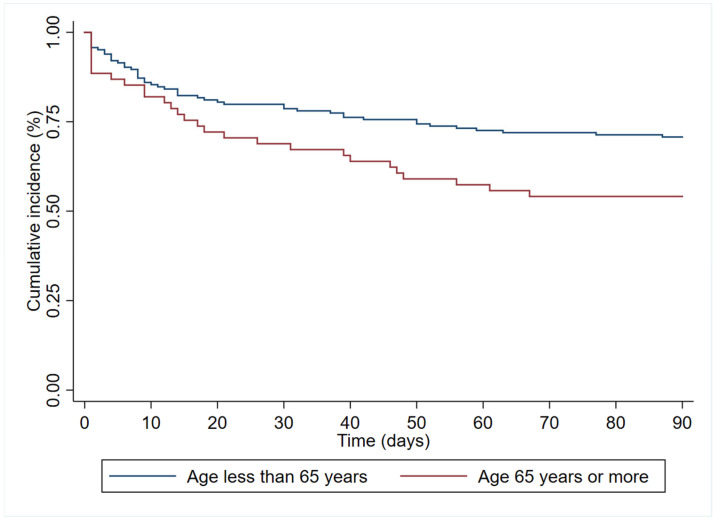
Ninety-day survival according to age.

**Figure 3 antibiotics-12-01130-f003:**
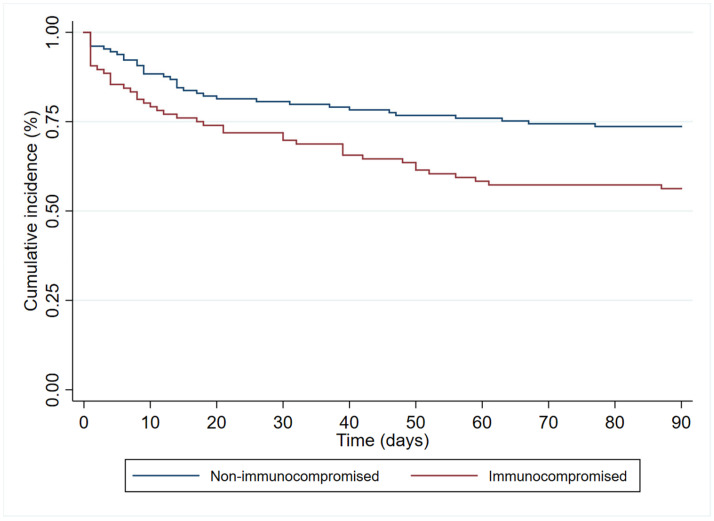
Ninety-day survival according to immunocompromise.

**Figure 4 antibiotics-12-01130-f004:**
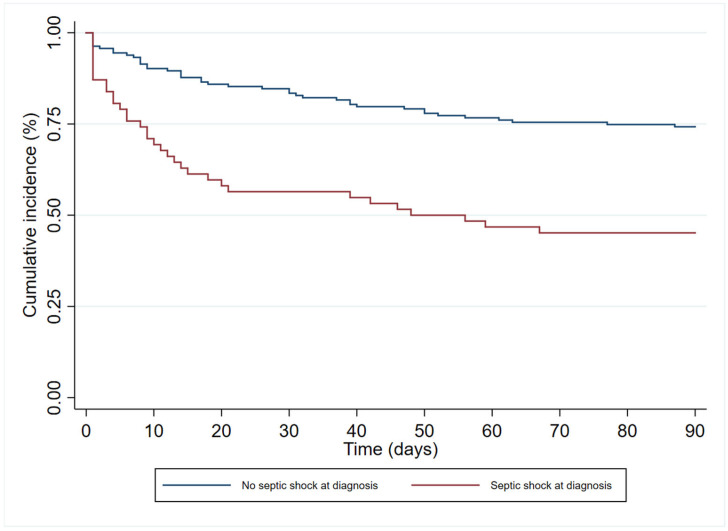
Ninety-day survival according to septic shock at diagnosis.

**Figure 5 antibiotics-12-01130-f005:**
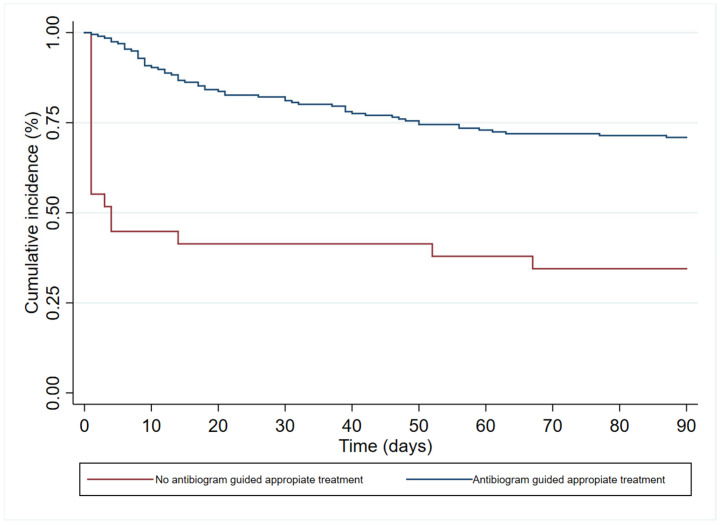
Ninety-day survival according to antibiogram guided appropriate treatment.

**Table 1 antibiotics-12-01130-t001:** Demographic and clinical characteristics.

Variable	Total n = 225 (100%)	Deaths at 90 Days n = 76 (33.8%)	Survivors at 90 Days n = 149 (66.2%)	*p*
Male, n (%)	145 (64.4)	53 (69.7)	92 (61.7)	0.236
Age—yr, median (IQR)	54 (40–66)	60 (47–70)	52 (38–63)	0.0055
Days from admission to diagnosis, median (IQR)	13 (4–28)	17 (5–32)	12 (4–27)	0.3421
Admission diagnosis, n (%)				0.468
Bacterial infection	125 (55.6)	42 (55.3)	83 (55.7)	
COVID-19	45 (20.0)	13 (17.1)	32 (21.5)	
Neoplasm	24 (10.7)	12 (15.8)	12 (8.1)	
Elective surgery	18 (8.0)	4 (5.3)	14 (9.4)	
Urgent surgery	8 (3.6)	3 (4.0)	5 (3.4)	
Other	5 (2.2)	2 (2.6)	3 (2.1)	
Patient location at diagnosis, n (%)				
ICU	81 (36.0)	37 (48.7)	44 (29.5)	0.005
Hospital ward	144 (64.0)	39 (51.3)	105 (70.5)	
Type of CR-GNB infection, n (%)				
Bloodstream infection	19 (8.4)	8 (10.5)	11 (7.4)	0.422
Respiratory tract infection	88 (39.1)	39 (51.3)	49 (32.9)	0.007
Intraabdominal infection	85 (37.8)	16 (21.1)	69 (46.3)	<0.001
Urinary tract infection	28 (12.5)	5 (6.6)	23 (15.5)	0.055
Bone and soft tissues infections	18 (8.0)	10 (13.2)	8 (5.4)	0.042
Others	4 (1.8)	0 (0)	4 (2.7)	0.303
Secondary bacteraemia	59 (26.2)	18 (23.7)	41 (31.5)	0.536
Comorbidities, n (%)				
Charlson index >3	88 (39.1)	41 (54.0)	47 (31.5)	0.001
COVID-19	56 (24.9)	17 (22.4)	39 (26.2)	0.532
Obesity	50 (22.4)	15 (19.7)	35 (23.8)	0.489
Diabetes	55 (24.4)	20 (26.3)	35 (23.5)	0.641
Heart disease	45 (20)	22 (29.0)	23 (15.4)	0.017
Hypertension	78 (34.7)	29 (38.2)	49 (32.9)	0.432
COPD	5 (2.2)	3 (4.0)	2 (1.3)	0.339
Immunosuppression	96 (42.7)	42 (55.3)	54 (36.2)	0.006
Iatrogenic bile duct injury	34 (15.1)	7 (9.3)	27 (18.1)	0.078
Liver cirrhosis	13 (5.8)	3 (4.0)	10 (6.7)	0.551
Renal replacement therapy for CKD	21 (9.3)	11 (14.5)	10 (6.7)	0.058
Cerebrovascular disease	7 (3.1)	4 (5.3)	3 (2.0)	0.230
Urological disorders	24 (10.7)	4 (5.3)	20 (13.4)	0.061
Tracheostomy carrier	31 (13.8)	9 (11.8)	22 (14.8)	0.547
Bacterial coinfection	120 (53.3)	41 (54.0)	79 (53.0)	0.895
Disease presentation severity, n (%)				
Mechanical ventilation at diagnosis	67 (29.8)	29 (38.2)	38 (25.5)	0.050
Septic shock at time of infection	62 (27.6)	34 (44.7)	28 (18.8)	<0.001

CKD: chronic kidney disease, CR-GNB: carbapenem-resistant Gram-negative bacilli, COPD: chronic obstructive pulmonary disease, CVD: cerebrovascular disease, IQR: interquartile range, ICU: intensive care unit.

**Table 2 antibiotics-12-01130-t002:** Secondary outcomes.

Outcome	Total n = 225 (100%)	Deaths at 90 Days n = 76 (33.8%)	Survivors at 90 Days n = 149 (66.2%)	*p*
Mechanical ventilation after diagnosis, n (%)	19 (8.4)	13 (17.1)	6 (4.0)	0.002
ICU stay after diagnosis, n (%)	25 (11.1)	16 (21.1)	9 (6.0)	0.001
Infection relapse, n (%)	41 (18.2)	8 (10.5)	33 (22.2)	0.033
Rehospitalisation for any cause, n (%) n = 224 *	41 (18.2)	8 (10.5)	33 (22.2)	0.033
C. difficile infection, n (%)	9 (4.0)	3 (4.0)	6 (4.0)	1.000
Antibiotic-related renal replacement therapy, n (%)	2 (0.9)	1 (1.3)	1 (0.7)	1.000
Acute kidney injury within the first 10 days of appropriate treatment, n (%) n = 181	40 (22.1)	21 (41.2)	19 (14.6)	<0.001
Days from diagnosis to death, median (IQR)	-	13 (4–38)	-	-

ICU: intensive care unit, IQR: interquartile range, * At the time of study cut-off, one patient was still hospitalised.

**Table 3 antibiotics-12-01130-t003:** Multivariate analysis for primary outcome.

Variable	aHR (CI 95%), *p*
Male gender	1.08 (0.64–1.83), 0.779
Age *	1.19 (1.00–1.83), 0.048
Bloodstream infection	1.55 (0.60–4.00), 0.364
Respiratory tract infection	1.25 (0.54–2.94), 0.599
Intraabdominal infection	0.57 (0.22–1.47), 0.245
Urinary tract infection	0.33 (0.10–1.10), 0.072
Bone and soft tissue infections	1.61 (0.59–4.38), 0.347
Obesity	0.84 (0.46–1.53), 0.564
Diabetes	0.97 (0.51–1.82), 0.919
Heart disease	1.51 (0.83–2.73), 0.179
Hypertension	0.69 (0.39–1.21), 0.193
Immunosuppression	1.84 (1.06–3.18), 0.030
Cirrhosis	1.37 (0.40–4.72), 0.618
Renal replacement therapy	1.65 (0.75–3.62), 0.212
Mechanical ventilation at diagnosis	1.26 (0.65–2.41), 0.493
Septic shock at time of infection	2.40 (1.41–4.08), 0.001
Non-fermenting Gram-negative bacilli	0.84 (0.50–1.42), 0.519
Antibiogram guided appropriate treatment	0.25 (0.14–0.46), <0.001

n = 223. aHR: adjusted hazard ratio, CI: confidence interval. * Adjusted hazard ratio for each one-year increase in age.

## Data Availability

The datasets used and/or analysed during the current study are available from the corresponding author (B.A.M.-G.) on reasonable request.

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
