# Peer review of "Predictors of Mortality in Patients with Infections Due to Carbapenem-Resistant Gram-Negative Bacteria"

_antibiotics, 2023, doi:10.3390/antibiotics12071130_

Round 1
Reviewer 1 Report
The authors report a retrospective monocentric experience on CR infectionsin hospitalized patients. The topic is interesting since we have the need of pointing on MDR infections, which is really a great problem nowaday. The data presentation is quite clear, however I believe some important information are missing. From a general point of view the authors do not considered screening for CRE wether it is perforemd or not, and if yes, how many patients were affected form CR-GNB after being colonized from CRE. Such information is quite important. Another general point regars the treatment: supplemental materials offer an overview on drugs used, however, although details on treatment are not expected according to the title, reporting more information about treatment, combined therapy for example, could add value to the paper, also because treatment resulted an important parameter influencing survival.
Detailed comments follow:
line 97 and 114: typing mistakes
METHODS and RESULTS: information on CRE colonization should be added
Label on figure 1: spelling mistake
RESULTS: line 143: the number of total patients should be specified: 145 out of 225?
line 144: "the time from admission to diagnosis": is not clear "diagnosis of infection" should be specified
line 148 and following: the presentation of percentages is not clear, the denominator should be always presented, or deleted referring to the table; also looking in the table 1 the sum of different infections gives a number which I cannot understand (242) does it mean that some patients have more than one type of infection? This should be more clearly expressed
DISCUSSION: the discussion generally needs to be revised: line 264 such statement does not add any interesting comment directely connected to the discussion; acomment on colonization and treatment should also be added in the discussion
REFERENCES: reference 5 is duplicated at number 8 and number 11; reference number 14 needs to be completed.
English language needs some spelling corrections, the paper is globally well written
Author Response
Dear Reviewer,
I hope this reply finds you well. On behalf of the authors, I would like to thank you for your thorough revision. Your comments and suggestions have undoubtedly improved the quality of our paper. Below, you will find a point-by-point response to your comments and suggestions.
- Comment: The authors report a retrospective monocentric experience on CR infections in hospitalized patients. The topic is interesting since we have the need of pointing on MDR infections, which is really a great problem nowadays. The data presentation is quite clear, however I believe some important information are missing. From a general point of view the authors do not considered screening for CRE whether it is performed or not, and if yes, how many patients were affected from CR-GNB after being colonized from CRE. Such information is quite important. Another general point regarding the treatment: supplemental materials offer an overview on drugs used, however, although details on treatment are not expected according to the title, reporting more information about treatment, combined therapy for example, could add value to the paper, also because treatment resulted an important parameter influencing survival.
- Answer: Thank you very much for your valuable comments. Infections due to MDR pathogens are indeed a global threat. In our centre, routine screening for CRE is not widely used as has been stated in lines 105-106 and 288-289. Fortunately, current research is underway to implement this useful strategy. We agree with your suggestion regarding the expansion of antibiotic treatment-related information. We have updated the manuscript in lines 206-209.
- Comment: line 97 and 114: typing mistakes
- Answer: the typing mistakes have been corrected.
- Comment: METHODS and RESULTS: information on CRE colonization should be added
- Answer: The fact that we do not routinely screen for CRE has been included in the manuscript in the lines 105-106 and has been acknowledged as a limitation in the lines 288-289.
- Comment: Label on figure 1: spelling mistake
- Answer: The mistake has been corrected.
- Comment: RESULTS: line 143: the number of total patients should be specified: 145 out of 225?
- Answer: The total number of patients has been specified
- Comment: line 144: "the time from admission to diagnosis": is not clear "diagnosis of infection" should be specified
- Answer: the information has been clarified accordingly.
- Comment: line 148 and following: the presentation of percentages is not clear, the denominator should be always presented, or deleted referring to the table; also looking in the table 1 the sum of different infections gives a number which I cannot understand (242) does it mean that some patients have more than one type of infection? This should be more clearly expressed.
- Answer: all denominators have been added. A total of 18 patients had more than one episodes of infection due to an CR organism. Only the first episode was considered for the multivariable analysis. The fact that 18 patients had more than one episode has been specified in line 155.
- Comment: DISCUSSION: the discussion generally needs to be revised: line 264 such statement does not add any interesting comment directly connected to the discussion; a comment on colonization and treatment should also be added in the discussion
- Answer: the statement has been removed from the discussion. Comments on colonization and treatment have been added to in lines 105-106, 206-209, 288-289 and 303-305.
- Comment: REFERENCES: reference 5 is duplicated at number 8 and number 11; reference number 14 needs to be completed.
- Answer: the mentioned issues have been corrected. The reference has been completed. CLSI. Performance Standards for Antimicrobial Susceptibility Testing. 30th CLSI supplement M100. Wayne, PA: Clinical and Laboratory Standards Institute, 2020.
- Comments on the Quality of English Language: English language needs some spelling corrections; the paper is globally well written.
- Answer: a thorough review to correct spelling and grammar mistakes has been done.
Thank you very much for your time and dedication.
Best regards,
Bernardo Alfonso Martínez Guerra
Reviewer 2 Report
Minor revisions:
- Key words are missing. Should the Authors complete this section?
- “Septic shock al time of infection” in tab. 1 should be change in “Septic shock AT time of infection”.
- Line 270: there is a repeated word.
Major revisions:
- Should the Authors clarify the “elevated laboratory markers” in line 110?
- Line 91-93: the reference for protocols or manufactures instruction is missing.
- The Authors should provide to add the denominators in the results section. It’s not clear if the analysis or data are referred to all cases involved, or some information are missing.
- Which is the difference between COVID at admission and COVID as comorbidities? Are the patients infected during the hospitalization? It appears confounding.
- How the Authors define the immunocompromised status?
- How many patients have been involved in multivariate model?
- The age cut-off should be detailed in table 3.
- Why did the Authors not consider the mortality rate at 30 days?
- Several clinical information was omitted in the multivariate analysis, such as length of stay, presence of multi-resistant microorganisms, ICU stay, although they represent all known factors that contribute to increased risk of mortality. The Authors did consider this feature in a preliminary univariate analysis?
Some errors are present in the text and, in general, the manuscript requires revision by an English language expert.
Author Response
REVIEWER 2
Dear Reviewer,
I hope this reply finds you well. On behalf of the authors, I would like to thank you for your profound revision. All your comments and suggestions are highly valuable to us, as they have undoubtedly improved the quality of our work. Below, you will find a point-by-point response to your comments and suggestions.
- Comment: Key words are missing. Should the Authors complete this section?
- Answer: We have fixed this issue and keywords have been added in line 49.
- Comment: “Septic shock al time of infection” in tab. 1 should be change in “Septic shock AT time of infection”.
- Answer: We have corrected this spelling mistake.
- Line 270: there is a repeated word.
- Answer: We have corrected this spelling mistake.
- Comment: Should the Authors clarify the “elevated laboratory markers” in line 110?
- Answer: we have clarified the aforementioned terms in line 116.
- Comment: Line 91-93: the reference for protocols or manufactures instruction is missing.
- Answer: a reference has been included,
- Comment: The Authors should provide to add the denominators in the results section. It’s not clear if the analysis or data are referred to all cases involved, or some information are missing.
- Answer: all denominators have been added.
- Which is the difference between COVID at admission and COVID as comorbidities? Are the patients infected during the hospitalization? It appears confounding.
- Answer: Although COVID-19 was one of the main reasons for hospital admission in 45 patients, SARS-CoV-2 was detected in 56 patients. In 11 patients, COVID-19 was detected but was not the main cause of hospital admission. A clearer language has been used in line 150-151.
- Comment: How the Authors define the immunocompromised status?
- Answer: Immunosuppression has been defined in the methods section. The definition is stated in lines 100-105.
- Comment: How many patients have been involved in multivariate model?
- Answer: a total of 223 patients were included in the multivariate model. The latter has been added in table 3.
- Comment: The age cut-off should be detailed in table 3.
- Answer: age was analyzed as a quantitative variable and, as such, adjusted HR represents an increase in risk for each one-year increase in age. The latter has been specified in table 3.
- Comment: Why did the Authors not consider the mortality rate at 30 days?
- Answer: We chose to study outcomes at 90 days to ensure a better understanding of the long-term impact of infections due to CR pathogens. We believe that studying outcomes at 90 days after diagnosis may improve the quality of the analysis, as we were able to document outcomes that could otherwise be unnoticed. Additionally, patients with infections due to CR organisms tend to have a higher frequency of non-infectious comorbidities that could impact the outcomes after a 30-day follow-up.
- Comment: Several clinical information was omitted in the multivariate analysis, such as length of stay, presence of multi-resistant microorganisms, ICU stay, although they represent all known factors that contribute to increased risk of mortality. The Authors did consider this feature in a preliminary univariate analysis?
- Answer: According to the prespecified analysis plan, secondary outcomes were not included in the predictive model of the primary outcome. Given that the multivariate analysis was performed to find independent associations with de primary outcomes, we chose to omit secondary outcomes (e.g. length of stay, infection relapse, invasive mechanical ventilation after the diagnosis of infection, ICU stay after diagnosis, need for additional source control, acute kidney injury, difficile infection, and all-cause hospital readmission) in the multivariable analysis. We believe that additional bias could influence the analysis if secondary outcomes were studied as independent risk factors for the primary outcomes. Additionally, Charlson was not included in the model since this variable includes age and comorbidities. COVID-19 was not included in the model due to interactions with obesity, diabetes, and upper respiratory tract infection. COPD was not included due to its interaction with heart disease. Iatrogenic bile duct injury presented interactions with abdominal sepsis was not included. Stroke interacted with diabetes mellitus was not included. Urological alterations and tracheostomy carriers interacted with the urinary tract and respiratory tract infection, respectively, and were not included. Bacterial co-infection was not included at the investigators' discretion as the failure of appropriate treatment is not expected. Secondary bacteremia was excluded because it was exclusive to primary bloodstream infections. Risk factors for acquiring BGNRC infections (e.g., prior chemotherapy, etc.) interacted with comorbidities (e.g. immunosuppression) and were not included. Laboratory parameters were not included as it is impossible to account for the variability of results over time and interactions with comorbidities. Carbapenemase phenotype and genotype were not included, as testing was not performed on the entire cohort.
- Comments on the Quality of English Language: Some errors are present in the text and, in general, the manuscript requires revision by an English language expert.
- Answer: a thorough review to correct spelling and grammar mistakes has been done.
Again, we would like to thank you very much for your time and dedication.
Kind regards,
Bernardo Alfonso Martínez Guerra
Reviewer 3 Report
Missing genotypic profiles of CR-GNB. Should be researched CR which is mediated by transferable carbapenemase-encoding genes. What was the rate of XDR and PDR among CR-GNB? Readers would like to know resistance mechanisms and case-fatality rate to risk CR-GNB infections.
Author Response
Dear Reviewer,
I hope this reply finds you well. On behalf of the authors, I would like to thank you for your revision. We fully agree with your suggestion. Below, you will find a point-by-point response your comment and suggestion.
- Comment: Missing genotypic profiles of CR-GNB. Should be researched CR which is mediated by transferable carbapenemase-encoding genes. What was the rate of XDR and PDR among CR-GNB? Readers would like to know resistance mechanisms and case-fatality rate to risk CR-GNB infections.
- Answer: given the retrospective nature of the study, we were unable to report complete genotypic profiles; therefore, specific carbapenem resistance mechanisms contributions to mortality could not be analysed. Unfortunately, in most of the reported studies, complete molecular data on resistance mechanisms is unavailable. Because of the latter, we believe that its absence in our study does not necessarily imply a major Fortunately, our centre has now implemented routine identification of resistance mechanisms. We believe that reporting the rate of XDR, defined as non-susceptibility to at least one agent in all but two or fewer antimicrobial categories, may not be entirely unbiased because we used clinical samples that were subjected to routine processes according to the sample origin (e.g. not all antibiotics were tested in all isolates). The latter is the reason why we preferred to report on the microbiological isolates and non-susceptibility frequency in the supplementary material.
Thank you very much for your time and dedication.
Kind regards,
Bernardo Alfonso Martínez Guerra
Reviewer 4 Report
The authors aimed to identify factors associated with 90-day all-cause mortality in patients with carbapenem-resistant Gram-negative bacteria (CR-GNB) infections. They determined presence of advanced age, immunosuppression, septic shock at diagnosis, and inappropriate treatment are associated with higher 90-day all-cause mortality.
Comments:
- The Introduction section is too short. Especially, the reasons why this study is conducted should be explained in more detail because studies determining association of mortality caused by carbapenem-resistant bacteria with different factors have been conducted already as authors stated themselves.
- Uniform terminology should be used in the whole manuscript and supplemental files - Enterobacterales should be used instead of Enterobacteriaceae
- In the Methods section the authors should describe isolates from which clinical samples were regarded as significant
- line 97 - "or mor" should be deleted
- Figure 1 - the heading of the figure should be corrected and brackets removed; suggestion for the heading: Figure 1. Flow diagram of patient inclusion
- Conclusion should be improved. The authors can use lines from 292 to 300 and include them in the Conclusion section
- line 270 - "in" is written twice
- Throughout the study (in the Introduction and in the end of the Discussion section) the authors are mentioning the importance of rapid diagnostics and implementation of antimicrobial stewardship program (ASP). However, they did not specifically explained the link between their results and rapid diagnostics and ASP.
Author Response
Dear Reviewer,
I hope this reply finds you well. On behalf of the authors, I would like to thank you for your thorough revision. Your comments and suggestions have undoubtedly improved the quality of our paper. Below, you will find a point-by-point response your comments and suggestions.
- Comment: The Introduction section is too short. Especially, the reasons why this study is conducted should be explained in more detail because studies determining association of mortality caused by carbapenem-resistant bacteria with different factors have been conducted already as authors stated themselves.
- Answer: the reasons why this study is conducted have been stated in lines 72-78.
- Comment: Uniform terminology should be used in the whole manuscript and supplemental files - Enterobacterales should be used instead of Enterobacteriaceae
- Answer: the terminology has been corrected.
- Comment: In the Methods section the authors should describe isolates from which clinical samples were regarded as significant.
- Answer: We screened all samples sent to the microbiology laboratory, regardless of its origin. The latter has been specified in the manuscript in lines 88-89. Significance was considered in accordance according to the attending team’s criterion, the latter has been specified in the methods section.
- Comment: line 97 - "or mor" should be deleted
- Answer: the mistake has been corrected.
- Comment: Figure 1 - the heading of the figure should be corrected and brackets removed; suggestion for the heading: Figure 1. Flow diagram of patient inclusion
- Answer: a correction has been implemented.
- Comment: Conclusion should be improved. The authors can use lines from 292 to 300 and include them in the Conclusion section.
- Answer: we have implemented the correction according to your comments.
- Comment: line 270 - "in" is written twice
- Answer: the mistake has been corrected.
- Comment: Throughout the study (in the Introduction and in the end of the Discussion section) the authors are mentioning the importance of rapid diagnostics and implementation of antimicrobial stewardship program (ASP). However, they did not specifically explain the link between their results and rapid diagnostics and ASP.
- Answer: the link between our findings and the need for rapid diagnostics has been highlighted in lines 309-312. Regarding the antimicrobial stewardship program, we believe our work may no be directly used to guide specific ASP strategies. Because of the latter, we chose not to focus the introduction and discussion in ASP. The paper has been modified accordingly.
Thank you very much for your time and dedication.
Kind regards,
Bernardo Alfonso Martínez Guerra
Round 2
Reviewer 2 Report
I'd like to thank the Authors for the detailed responses provided.
All point were reviewed in the manuscript according to my suggestion. I would suggest just a couple of things before publication:
- 1) The Authors should add in the methods or discussion sections a clarification regarding the choice of 90-day outcome vs. 30-day mortality.
- 2) I kindly recommend a revision of English language.
Best regards
The text is not very fluent and some terms are repeated. Further revision of the text would be needed.
Author Response
Cover letter- Reply round 2
Bernardo A Martinez-Guerra, M.D., MSc.
Department of Infectious Diseases
Instituto Nacional de Ciencias Médicas y Nutrición Salvador Zubirán
15 Vasco de Quiroga, Belisario Domínguez Sección XVI, Tlalpan
Mexico City, Mexico. 14080
Tel, +5255-5487-0900. Ext. 5869
beramg@gmail.com
bernardo.martinezg@incmnsz.mx
June 18, 2023
Dear Jean Lu
Assigned Editor
Antibiotics
We hope this letter finds you well. We wish to resubmit an original article entitled “Predictors of Mortality in Patients with Infections due to Carbapenem-Resistant Gram-Negative Bacteria” for consideration to be published by Antibiotics. We gladly received the reviewers’ suggestions and comments. On behalf of the authors, I would again like to thank you and the reviewers for your time and dedication. We insist that all of the reviewers’ comments and suggestions have undoubtedly improved the quality of our article. Below, you will find a point-by-point response to each of the comments and suggestions.
We confirm that this work is original and has not been published, nor is it currently under consideration for publication elsewhere. All authors have reviewed and agreed with the contents of the manuscript. Additionally, we have no conflicts of interest to disclose.
Thank you for your consideration of this manuscript.
Kind regards,
Bernardo A Martinez-Guerra, M.D., MSc
Dear Reviewer,
I hope this reply finds you well. On behalf of the authors, I would again like to thank you for your thorough revision. As ever, your comments and suggestions are highly valuable to us. Below, you will find a point-by-point response to your comments and suggestions.
Comment: The Authors should add in the methods or discussion sections a clarification regarding the choice of 90-day outcome vs. 30-day mortality.
Answer: a clarification was added in lines 307-311.
Comment: I kindly recommend a revision of English language.
Answer: a revision has been done to increase text fluency and to avoid using repeated terms unnecessarily.
Kind regards,
Bernardo A Martinez-Guerra